# A Mid-Infrared Multifunctional Optical Device Based on Fiber Integrated Metasurfaces

**DOI:** 10.3390/nano13172440

**Published:** 2023-08-28

**Authors:** Weikang Yao, Qilin Zhou, Chonglu Jing, Ai Zhou

**Affiliations:** 1The School of Materials Science and Engineering, Wuhan University of Technology, Wuhan 430074, China; 2National Engineering Research Center of Fiber Optic Sensing Technology and Networks, Wuhan University of Technology, Wuhan 430070, China

**Keywords:** metasurface, Fano resonance, mid-infrared, optical device, refractive index sensor, optical filter

## Abstract

A metasurface is a two-dimensional structure with a subwavelength thickness that can be used to control electromagnetic waves. The integration of optical fibers and metasurfaces has received much attention in recent years. This integrated device has high flexibility and versatility. We propose an optical device based on fiber-integrated metasurfaces in the mid-infrared, which uses a hollow core anti-resonant fiber (HC-ARF) to confine light transmission in an air core. The integrated bilayer metasurfaces at the fiber end face can achieve transmissive modulation of the optical field emitted from the HC-ARF, and the Fano resonance excited by the metasurface can also be used to achieve refractive index (RI) sensing with high sensitivity and high figure of merit (FOM) in the mid-infrared band. In addition, we introduce a polydimethylsiloxane (PDMS) layer between the two metasurfaces; thus, we can achieve tunable function through temperature. This provides an integrated fiber multifunctional optical device in the mid-infrared band, which is expected to play an important role in the fields of high-power mid-infrared lasers, mid-infrared laser biomedicine, and gas trace detection.

## 1. Introduction

In recent years, there has been tremendous innovation in optics and photonics with the advent of the metasurface, a periodic arrangement structure of multiple units on a two-dimensional plane that can be used to tune electromagnetic waves [1,2]. Various metasurfaces have been designed on subwavelength dimension cells to modulate the phase, polarization mode, and propagation pattern of electromagnetic waves. Optical functional devices with high regulatory freedom and thin device thickness can be prepared, such as a polarization-dependent tunable hyperbolic microcavity [3] and a tunable achromatic waveplate [4]. Studying from the metasurface structure [5,6,7,8,9], it can be categorized into raised structural metasurfaces and hole-type metasurfaces. Compared with raised structural metasurfaces, hole-type metasurfaces such as photonic crystal slabs (PCSs) are less difficult to process and more stable. Different PCSs were developed for optical filtering and sensing [10,11,12,13,14]. Due to the insufficient performance of single-layer PCSs, to improve their filtering and sensing performance, double-layer PCSs [15,16,17,18,19] have been designed to achieve a very high Q-factor by controlling the coupling between the plates. Most of these double-layer PCSs concentrate on the electric field between two PCSs, and their light–matter interaction is not enough, resulting in low refractive index sensitivity [17,19]. This has restricted the development of efficient filtering and trace gas measurement.

Generally, a conventional metasurface is an independent device that is separated and poorly integrated, thus lacking wholeness and systematicity in practical applications. This greatly limits the application of metasurface devices [20,21,22]. Since silica fiber was proposed [23], optical fiber has been a mature optical transmission medium, and its excellent optical transmission performance and anti-electromagnetic interference properties have led to the rapid development of optical fiber sensing [24] and optical fiber communication [25]. Meanwhile, metasurface and optical fiber integration has also attracted great interest from researchers in the past decade [26]. Various optoelectronic functional materials and metasurface devices have been integrated into optical fibers, on the side of optical fibers, or on the end face of optical fibers. It has a significant impact in the fields of device integration, biosensing, information processing, and environmental monitoring [27,28,29]. The mid-infrared band with wavelengths of 2 μm~20 μm not only contains important windows of atmospheric transparency but also covers the characteristic spectral lines of numerous atoms and molecules. Hollow core anti-resonant fiber (HC-ARF) is a new type of optical fiber, which uses an anti-resonance effect to confine light transmission in the air hole in the mid-infrared band. Combining HC-ARF and metasurfaces into integrated devices is extremely important for many fields [26], such as space optical communication, atmospheric environment monitoring, and biomedicine.

In this paper, we propose a mid-infrared multifunctional optical device based on HC-ARF integrated metasurfaces for efficient transmission, filtering, and sensing in the mid-infrared band. The HC-ARF, which confines light transmitting in its air core, breaks the transmission limitations of conventional fiber optic materials in the mid-infrared band. The metasurface is composed of double-layer PCS, in which a layer of PDMS is introduced in the middle of two PCS layers. The employment of the PDMS makes the electric field in the PCSs be concentrated and transferred to both sides of the metasurface, rather than concentrated between the PCSs. This is because the RI difference between the PDMS and metasurface material is much smaller than that between the air and metasurface. As a result, we got a high interaction between light and matter and therefore a high external RI sensitivity. In addition, PDMS is high polymer material with a high thermo-optical coefficient and thermal expansion coefficient, and the wavelength of the proposed device can be tuned by changing the temperature of the PCSs, which can eliminate the errors caused by experimental processing. Firstly, we used COMSOL Multiphysics^®^ 5.6 software to calculate the two-dimensional HC-ARF end face to know its low limiting loss and good single-mode transmission performance. Then, we designed a bilayer silicon nitride metasurface with PDMS in the middle layer and simulated one of the structural units of the metasurface in three dimensions. As a result of the double Fano resonance, two transmission troughs are generated, and their Q-factors can reach 4929 and 733, with RI sensitivities of 2074 nm/RIU (RIU is the refractive index unit) and 955 nm/RIU and FOM values of 3050 and 108.5, respectively. We also obtain the cause of its Fano resonance generation and the contribution of each polariton generation by multipole analysis. By simulating the effects of its key parameters on the transmission spectrum, we know that its Q-factor can continue to be improved, but at the same time, it needs better processing means, which is something we need to consider in detail in the actual fabrication.

## 2. Structure Design and Simulation Setup

We used an HC-ARF as the transmission medium to achieve light transmission in the mid-infrared band and placed a metasurface on its end face. The 3D schematic diagram of the proposed mid-infrared multifunctional optical device is shown in Figure 1a, which consists of a hollow core anti-resonant fiber (HC-ARF) and a bilayer of silicon nitride (Si_3_N_4_) metasurface. Figure 1b shows the cross-section of HC-ARF, which consists of a quartz cladding, six glass tubes, and an air core. The fiber core diameter of the HC-ARF is defined as *D*, the inner diameter of the cladding tube as *d,* and the thickness of the glass wall as *t.* The structure of the metasurface is shown in Figure 1c,d. The bottom and top layers are silicon nitride films, the middle layer is filled with low refractive index PDMS, and finally, the periodic air holes are machined to obtain a bilayer silicon nitride metasurface. The thickness of the silicon nitride film is *h*_1_, the thickness of the PDMS film is *h*_2_, the period of the air hole is *P*, and its radius is *r*.

For the HC-ARF, we calculated the mode analysis of its two-dimensional (2D) cross-section using the commercial finite element simulation software COMSOL Multiphysics based on the finite element method (FEM). Using the boundary condition of a perfect matching layer (PML), the mode field distribution and transmission coefficient of HC-ARF can be analyzed to obtain the main characteristic parameters of transmission modes in optical fiber, such as the mode effective refractive index, confinement loss (CL), high order mode extinction ratio (HOMER), etc. The material of the fiber is silica, and its refractive index is set to 1.4449, setting the key structural parameters of HC-ARF as *D* = 33 μm, *d* = 22.8 μm, and *t* = 1 μm.

For the bilayer Si_3_N_4_ metasurfaces, we can obtain its s-parameters, transmittance, electric and magnetic fields, and other parameters by calculating the unit cell through a three-dimensional (3D) simulation from COMSOL Multiphysics using periodic boundary conditions (PBCs). In addition, the simulation can be completed quickly and accurately by correctly setting boundary conditions, including perfect matching layers, ports, and so on. We used data from the COMSOL material library for Si_3_N_4_ and set the refractive index of PDMS to 1.41, setting the key structural parameters of the bilayer Si_3_N_4_ metasurface as *h*_1_ = 220 nm, *h*_2_ = 220 nm, *P* = 2800 nm, and *r* = 400 nm.

Here, the proposed fiber-integrated metasurface is a complicated device. Samples can be fabricated in the following possible procedure. Firstly, we can deposit two layers of Si_3_N_4_ on the Si substrate by chemical vapor deposition (CVD); then, we can spin coat PDMS on the Si_3_N_4_ by using the transfer method to create a three-layer structure. Then, we can fabricate holes by electron-beam lithography (EBL) [30]. Next, by applying a certain adhesive to the end face of the optical fiber, transferring the metasurface structure to the end face [28], and, finally, releasing the metasurface from the SI substrate using potassium hydroxide (KOH), the structure can be successfully processed.

## 3. Simulation Results and Discussion

### 3.1. Theoretical Analysis and Simulation of HC-ARF

The light-guiding mechanism of HC-ARF is based on the anti-resonant reflecting optical waveguide (ARROW) model, which is a flat-plate waveguide theory. Initially, in 2002, Litchinitser et al. used this theory to explain the principle of light conduction in Kagome fibers [31]. The principle of anti-resonant light-guiding is shown in Figure 2, when a beam of light is incident into an air core, the surface of the glass wall is approximated as a grazing incidence, and part of the light is emitted and returned to the core; the other part of the light is refracted and enters the glass wall, forming a Fabry–Perot-like resonant cavity in the glass wall. The wavelength of the incident light and the thickness of the glass wall determine the resonance conditions and anti-resonance conditions of the resonant cavity, and the light that meets the resonance conditions is leaked out, while the light that does not meet the resonance conditions through the anti-resonance effect back into the fiber core propagation, through this principle can be limited to the light propagation in the air fiber core. Snell’s law and the Fabry–Perot cavity principle can be deduced from the antiresonance conditions as follows:(1)$$\lambda =\frac{2t\sqrt{n_{1}{}^{2}-n_{0}{}^{2}}}{m-0.5}$$
where λ is the incident wavelength, t is the thickness of the glass wall, $n_{1}$ is the refractive index of the glass wall, $n_{0}$ is the refractive index of air, and m is the resonance order. When the anti-resonance condition is satisfied, the reflection from the glass wall is the maximum and the transmission is the minimum, and most of the light is reflected into the fiber core, so that the light is transmitted through the air hole.

We designed a single-layer circular tube HC-ARF operating in the mid-infrared band, a two-dimensional cross-section is shown in Figure 1b, and its mode characteristics and single-mode transmission performance are analyzed using COMSOL software. The material of the fiber is silica, and its refractive index is set to 1.4449. Its structure parameters are *D* = 33 μm, *D* = 22.8 μm, and *t* = 1 μm. Through the mode analysis, we can obtain the variation of its effective refractive index with wavelengths for different modes, which is the dispersion curve. The confinement loss is the main source of loss for HC-ARF in a straight state [32]. The confinement loss of the fiber can be calculated from the imaginary part of the effective refractive index as follows:(2)Confinementloss=20ln10⋅2πλ⋅Im(neff)
where λ is the incident wavelength, and Im(neff) is the imaginary part of the effective refractive index.

We obtained the fundamental mode and several typical higher-order modes at 3140 nm by the mode analysis, including the LP_01_ (HE_11_) mode; LP_02_ (HE_12_) mode; LP_11_ (TM_01_, TE_01_, and HE_21_) mode; and LP_21_ (HE_31_ and HE_11_) mode. The mode field distribution is shown in Figure 3.

In addition, we calculated the effective refractive index curves with wavelengths from 2500 nm to 3500 nm, as shown in Figure 4a. According to Equation (1), we can calculate the confinement loss of HC-ARF, and we can get the confinement loss curve with the wavelength for each mode, as shown in Figure 4b. Its inset shows the loss spectrum of quartz material, and the data are from Heraeus [33]. The fundamental mode loss of HC-ARF can reach as low as 0.03 dB/m at 2900 nm, so HC-ARF can transmit at a very low loss in the mid-infrared band, while the loss of the quartz material reaches 54 dB/m at 3000 nm, and the loss rises sharply at longer wavelengths. Therefore, conventional single-mode fibers are not suitable for transmission in the mid-infrared band.

In most optical fibers, a single mode is required for efficient transmission, but in HC-ARF, due to the nature of leaking modes, we can only achieve single-mode-like transmissions by suppressing the transmissions of other higher-order modes and by making the loss of other higher-order modes much larger than the loss of the fundamental mode; the higher-order modes quickly disappear during transmission, leaving only single-mode transmission in the fiber. To measure the single-mode performance, HOMER is defined as the ratio of the lowest loss of the higher-order mode to the loss of the fundamental mode. Therefore, HOMER is currently an important parameter to measure its single-mode transmission performance. We have calculated several common low-loss high-order modes of HC-ARF, and HOMER is up to more than 100, which can be considered ideal for single-mode-like transmissions, according to the current research proof [34].

### 3.2. Theoretical Analysis and Simulation of Si_3_N_4_ Metasurface

Fano resonance is generated by the coupling between two modes—namely, the coupling between the continuous and the discrete modes. There are three common mechanisms for achieving Fano resonance: one is interference cancellation between two distinct mode resonators, and another is interference cancellation between a bright mode and a dark mode resonator. Here, a resonant mode that can be directly excited by incident light is called a bright mode, while a mode that cannot be directly excited is called a dark mode. The third mechanism is interference cancellation between a bright mode and a trapped mode. However, trapped modes often require other ways of excitation, such as breaking the structural symmetry or using the oblique incidence of electromagnetic waves. We can first use the Lorentz resonance model to analyze the interactions between the modes from an academic perspective [35,36]:(3)$$\left\{\begin{array}{l} {\ddot{x}}_{1}\left(t\right)+\gamma_{1}{\dot{x}}_{1}\left(t\right)+\omega_{1}x_{1}\left(t\right)+kx_{2}\left(t\right)=E\left(t\right)\\ {\ddot{x}}_{2}\left(t\right)+\gamma_{2}{\dot{x}}_{2}\left(t\right)+\omega_{2}x_{2}\left(t\right)+kx_{1}\left(t\right)=0 \end{array}\right.$$
where ${\ddot{x}}_{1}\left(t\right)$ and ${\ddot{x}}_{2}\left(t\right)$ represent two different resonance modes, $\gamma_{1}$ and $\gamma_{2}$ represent the damping coefficients of two modes, $\omega_{1}$ and $\omega_{2}$ represent their intrinsic resonant frequencies, and k represents the coupling coefficient. The external electric field affects the model, and we performed Fourier transform on the formula:(4)$$\left(\begin{array}{cc} \omega_{1}^{2}-\omega^{2}+i\omega \gamma_{1} & K\\ K & \omega_{2}^{2}-\omega^{2}+i\omega \gamma_{2} \end{array}\right)\left(\begin{array}{l} x_{1}\\ x_{2} \end{array}\right)=\left(\begin{array}{c} E(t)\\ 0 \end{array}\right)$$

Then, the amplitude A can be expressed as:(5)$$A=E(t)\frac{\omega_{2}^{2}-\omega^{2}+i\omega \gamma_{2}}{(\omega_{1}^{2}-\omega^{2}+i\omega \gamma_{1})(\omega_{2}^{2}-\omega^{2}+i\omega \gamma_{2})-k^{2}}$$

The transmission spectrum T is expressed as:(6)$$T=1-\left|\frac{A}{E(t)}\right|=1-{\left[\frac{\omega_{2}^{2}-\omega^{2}+i\omega \gamma_{2}}{(\omega_{1}^{2}-\omega^{2}+i\omega \gamma_{1})(\omega_{2}^{2}-\omega^{2}+i\omega \gamma_{2})-k^{2}}\right]}^{2}$$

From the above equations, it can be deduced that the transmission spectrum and amplitude of the system are related to the coupling coefficient k between the individual modes. Similarly, in Fano resonance, the asymmetry can be expressed by the following equation:(7)$$I\propto \frac{{\left(q\gamma +\omega -\omega_{0}\right)}^{2}}{{\left(\omega -\omega_{0}\right)}^{2}+\gamma^{2}}$$
where $\omega_{0}$ and $\gamma_{}$ are the resonant frequency and bandwidth, and q is the asymmetric coefficient. When q=1, the intensities of the discrete and continuous propagation modes are comparable, and the coupling is the strongest, resulting in an asymmetric line-shaped resonance peak. When q=0 or q=∞, there is a significant difference in intensity between the continuous and discrete modes, and their coupling only exhibits the propagation characteristics of the mode with the higher intensity. This can be analyzed from an academic perspective.

#### 3.2.1. Single-layer Si_3_N_4_ Metasurface

We first used a single layer of silicon nitride metasurface with periodically arranged air holes. We set its parameters as *h* = 250 nm, *P* = 2800 nm, and *r* = 400 nm. Periodic boundary conditions were set in the *x*-direction and *y*-direction to simulate a single metasurface consisting of numerous structural units. A perfect matching layer was set in the *z*-direction, and ports were set as the input and output for electromagnetic waves. At the input port, the components of the fundamental mode electric field of the HC-ARF were set as the input components. This allowed for the integrated simulation of the two-dimensional fiber and three-dimensional metasurface, as shown in Figure 5. Due to the periodically arranged air holes in the metasurface, the light output from the anti-resonant fiber was vertically incident on the metasurface, effectively exciting Fano resonance, as shown in Figure 6a with a sharp Fano peak. We can use the classical Fano formula [37] to fit the transmission spectrum, as follows:(8)$$T_{\mathrm{Fano}}={\left|a_{1}+ia_{2}+\frac{b}{\omega -\omega_{0}+i\gamma }\right|}^{2}$$
where $a_{1}$, $a_{2}$, and b are constant real numbers, $\omega_{0}$ is the central resonant frequency, and *γ* is the total damping rate of the resonator. This formula can also be used to accurately extract the *Q*-factor:(9)$$Q=\frac{\omega_{0}}{2\gamma }=\frac{2\pi c}{\lambda 2\gamma }$$
where c is the velocity of light, according to Equations (8) and (9). The simulation results are in agreement with the fitting results. This asymmetric line shape results from the destructive interference between the continuous and discrete states near the resonant center frequency. The red dotted line in Figure 6a depicts a fitted curve with detailed parameters: *a*_1_ = 1.221, *a*_2_ = −0.3614, *b* = 109.26, and *Q* = 835.24.

#### 3.2.2. Bilayer Si_3_N_4_ Metasurface

To obtain better filtering and sensing performance, we superimposed two silicon nitride films, bonded by a PDMS layer in the middle, and then formed a bilayer silicon nitride metasurface by periodically opening holes, thus constructing a Fabry–Perot (FP) cavity-like structure. Compared with the single-layer silicon nitride metasurface, a new mode can be stimulated, such as mode I in Figure 7a, which has a sharper peak and a higher Q-factor compared with mode 2. One resonance dip appears at 3133.3 nm (mode I) and another at 3572 nm (mode II). Both show typical Fano peaks. Figure 7b,c fit the two resonance peaks, respectively, as shown by the red dashed lines, with Q-factors of 4929 and 733, respectively. However, the thickness of PDMS is smaller than the incident wavelength and therefore cannot be explained using the theory of FP cavities but, rather, the Fano interference principle. There will be a strong field binding between the two layers of the metasurface, and its surface is sensitive to environmental changes and can be used as a high-sensitivity sensor. With a strong local field enhancement in the near field, the high thermo-optical coefficient of PDMS and its high sensitivity to the external environment can be used to change the near field coupling between the two layers of the metasurface to obtain a very high Q-factor, and the filtering performance can be greatly improved, and high-sensitivity refractive index sensing can also be achieved.

The electric field mode distribution at the resonance alone cannot be observed to accurately determine the mode, because there may be more than one mode at the resonance, and the strong mode will override the weak mode. Therefore, to evaluate the mechanism of both modes’ generation, the far-field radiation of Fano resonance is decomposed into the contributions of multiple components to further discuss their characteristics. These include electric dipole (ED), magnetic dipole (MD), toroidal dipole (TD), electric quadrupole (EQ), and magnetic quadrupole (MQ). We use the multipole decomposition method to calculate the multipole far-field scattering power:(10)$$\mathrm{ED}:\;\;\;\;\;\;\vec{P}=\frac{1}{i\omega }\int jd^{3}r,$$
(11)$$\mathrm{MD}:\;\;\;\;\;\;\vec{M}=\frac{1}{2c}\int (r\times j)d^{3}r,$$
(12)$$\mathrm{TD}:\;\;\;\;\;\;\vec{T}=\frac{1}{10c}\int [(r\cdot j)r-2r^{2}j]d^{3}r,$$
(13)$$\mathrm{EQ}:\;\;\;\;\;\;Q_{\alpha ,\beta }^{\left(e\right)}=\frac{1}{i2\omega }\int [r_{\alpha }j_{\beta }+r_{\beta }j_{\alpha }-\frac{2}{3}(r\cdot j)]d^{3}r,$$
(14)$$\mathrm{MQ}:\;\;\;\;\;\;Q_{\alpha ,\beta }^{\left(m\right)}=\frac{1}{3c}\int [{\left(r\times j\right)}_{\alpha }r_{\beta }+{\left(r\times j\right)}_{\beta }r_{\alpha }]d^{3}r,$$
where *c* is the speed of light in vacuum, *j* is the induced volume current density, ω is the angular frequency, r is the position vector, and *α*, *β* = *x*, *y*, *z.* The scattering power of the induced multipole moments contributing to the far field response can be written as:(15)$$I=\frac{2\omega^{4}}{3c^{3}}{\left|\vec{P}\right|}^{2}+\frac{2\omega^{4}}{3c^{3}}{\left|\vec{M}\right|}^{2}+\frac{2\omega^{6}}{3c^{5}}{\left|\vec{T}\right|}^{2}+\frac{\omega^{6}}{5c^{5}}{\sum \left|Q_{\alpha ,\beta }^{\left(e\right)}\right|}^{2}+\frac{\omega^{6}}{40c^{5}}{\sum \left|Q_{\alpha ,\beta }^{\left(m\right)}\right|}^{2}+o(\omega ).$$

By a multipole analysis, the far-field radiation intensity and distribution of the two mode multipoles are shown in Figure 8. It can be seen that mode I is mainly dominated by MD and EQ in Figure 8a, the magnetic dipole can be considered as the “bright mode” of radiation and EQ as the “dark mode”. The destructive interference between them excites the Fano resonance. Therefore, properties such as local field enhancement caused by MD and EQ are among the most important elements to obtain a high Q and high sensitivity. In Figure 8b, it can be seen that mode 2 is dominated by TD and MQ. Unlike electric and magnetic dipoles, toroidal dipoles originate from an electric field oriented along the meridian of the ring surface and a magnetic field oriented along the equatorial line or from a vortex configuration in which the heads and tails of magnetic dipoles are connected [38].

We investigated the effects of several important structural parameters on the transmission spectrum, including the thickness of the silicon nitride film *h*_1_, the thickness of the PDMS film *h*_2_, and the radius of the air holes *r*. The spacing between the two metasurfaces is an important parameter for mode coupling. According to the electric field distribution of the two modes, it is known that the electric field of mode I is distributed between and outside the two metasurfaces, while mode II is mainly distributed in the air holes of the metasurfaces. As shown in Figure 9a–c, for the variations of the thickness between the metasurfaces, dip1 has a large offset, while dip2 has no offset. For the variations of the hole radius *r*, dip1 will have no offset, while dip2 will be affected by the hole size more. As shown in Figure 9d,e, the resonant wavelength and Q-factor of mode I can be obtained as a function of the change in film thickness.

Most of the area of the metasurfaces is exposed to air, and its hole structure and bilayer structure also enhance the interaction between light and matter. For mode I, assuming that the simulation is an air hole between the two layers of the metasurfaces, the electric field is mainly concentrated between the two plates, and the coupling of gas molecules occurring during the electric field distribution is not strong enough. We add a layer of PDMS film in the middle layer of the two metasurfaces, which not only plays the role of connecting the two metasurfaces but also makes the refractive index difference between the material in the cavity and the metasurface material decrease and concentrates the electric field distribution to the outer side of the two metasurfaces. This enhances the interaction between the gas molecules in the air and the metasurface so that the destructive interference between the modes will change with the refractive index of the external gas, which is one of the important factors that make this structure a highly sensitive refractive index sensor. We can use refractive index sensitivity and FOM to define the performance of the sensor. By setting the background refractive index, *n* from 1 to 1.04, in COMSOL Multiphysics, the process of gradually increasing the concentration of a particular gas in the air can be simulated, and the simulation results are shown in Figure 10a. It can be learned that, when the gas concentration gradually increases, the ambient refractive index gradually increases, and the transmission spectrum is red-shifted, where the change of mode I is more obvious. Next, we can calculate the refractive index sensitivity, with the following expression [39]:(16)$$S_{\lambda }=\frac{\Delta \lambda }{\Delta n}(\mathrm{nm}/\mathrm{RIU})$$
where Δλ represents the offset of the resonant wavelength, and Δn represents the change in the background refractive index. FOM is defined as follows [6]:(17)$$FOM=\frac{S_{\lambda }}{FWHM}({\mathrm{RIU}}^{-1})$$
where *FWHM* is the full width at the half-maximum of the resonant wavelength.

As shown in Figure 10b,c, the RI sensitivity of mode I is as high as 2074 nm/RIU, and its FWHM is 0.68 nm, so its FOM value is 3050 RIU^−1^. The refractive index sensitivity of mode II is 955 nm/RIU, and its FWHM is 8.8 nm, so its FOM value is 108.5 RIU^−1^. Mode I has better sensing characteristics. It has a strong application prospect in measuring the gas content in the air. Compared with other dielectric metasurfaces, we have a higher refractive index sensitivity with a high FOM value. Metal metasurfaces have a high refractive index sensitivity, but metallic materials themselves can have strong ohmic losses. Our proposed dielectric metasurface has not only low loss but also a high refractive index sensitivity and FOM value. Therefore, this structure has a high potential for filtering and refractive index sensing.

In addition, PDMS is a highly polymer material with a high thermo-optical coefficient and thermal expansion coefficient, and the wavelength of the proposed device can be tuned by changing the temperature of the environment, because the thickness and RI of PDMS vary with the temperature. Thus, it can be used to eliminate the errors caused by experimental processing. We can calculate the RI and thickness of PDMS with the following expression [40]:(18)$$H_{PDMS}=300\times 10^{-6}\times H_{0}\times (T-20)+H_{0}$$
(19)$$n_{PDMS}=-4.5\times 10^{-4}\times (T-20)+1.41$$
where $H_{PDMS}$ represents the thickness of PDMS under different temperature environments, $H_{0}$ represents the initial thickness of PDMS, T represents the temperature of the environment and $n_{PDMS}$ represents RI of PDMS under different temperature environments.

As shown in Figure 11, it can adjust the position of the filtered resonant peak by controlling the temperature. This tunability plays a significant role in precise filtering, eliminating machining errors, and other aspects.

## 4. Conclusions

In summary, we propose an integrated fiber and metasurface device in the mid-infrared band, which achieves low-loss transmission in the mid-infrared band down to 0.03 dB/m at 2900 nm with good single-mode transmission performance. The double Fano resonance is excited by the bilayer metasurface structure, and the Q-factor is greatly enhanced compared with that of the single-layer super surface. The multipole analysis shows that mode I is excited due to the destructive interference between MD and EQ. With the parameters we designed, mode I obtained a Q-factor of 4929, refractive index sensitivity of 2074 nm/RIU, and a 3050 FOM value. In addition, we found that its Q-factor can be enhanced by adjusting the structural parameters by studying the effect of the structural parameters on the transmission spectrum. Its high Q-factor, high sensitivity, and high FOM value are potentially valuable for tunable filters, sensors, and mid-infrared integrated devices.

## Figures and Tables

**Figure 1 nanomaterials-13-02440-f001:**
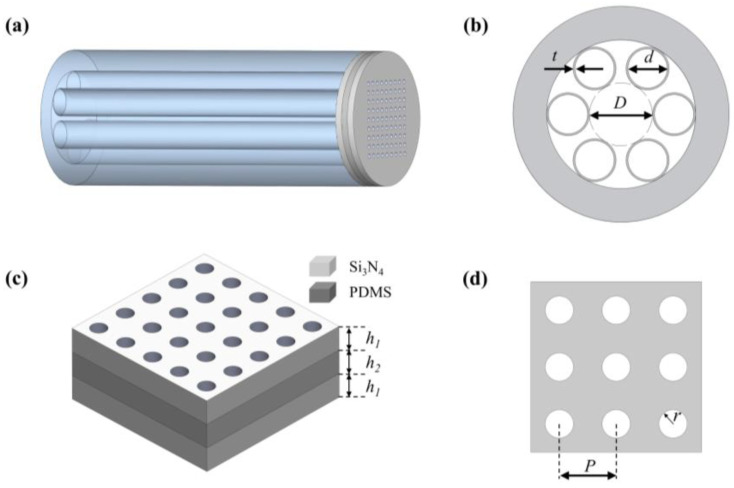
(**a**) Schematic diagram of the proposed mid-infrared multifunctional optical device. (**b**) The schematic diagram of the HC-ARF cross-section. (**c**) Schematic diagram of a double-layer silicon nitride metasurface filter. (**d**) Key parameters of a silicon nitride metasurface.

**Figure 2 nanomaterials-13-02440-f002:**
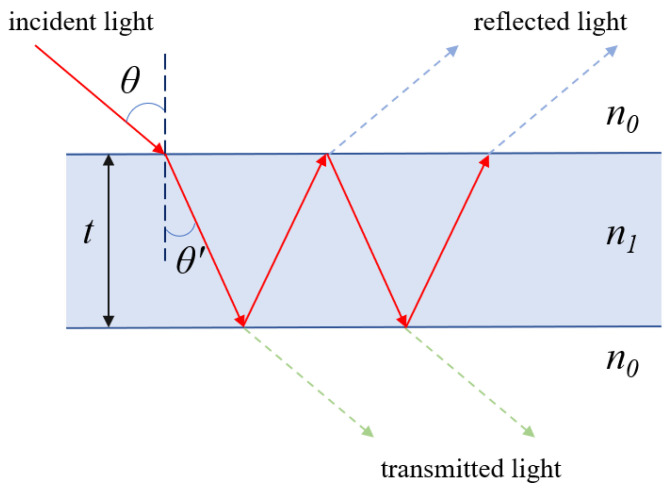
Schematic diagram of the ARROW model.

**Figure 3 nanomaterials-13-02440-f003:**
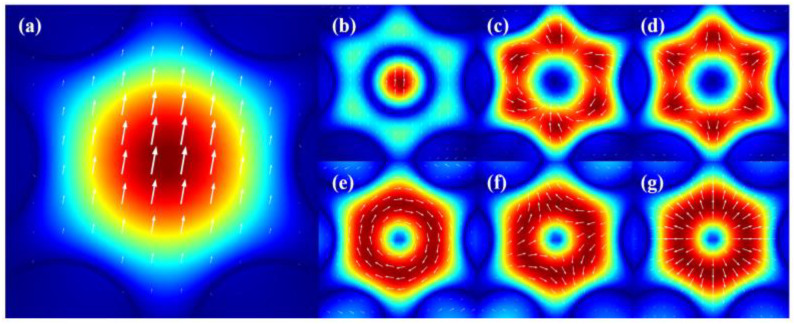
Mode field distribution of different modes. (**a**) HE_11_ mode. (**b**) HE_12_ mode. (**c**) EH_11_ mode. (**d**) HE_31_ mode. (**e**) TE_01_ mode. (**f**) HE_21_ mode. (**g**) TM_01_ mode.

**Figure 4 nanomaterials-13-02440-f004:**
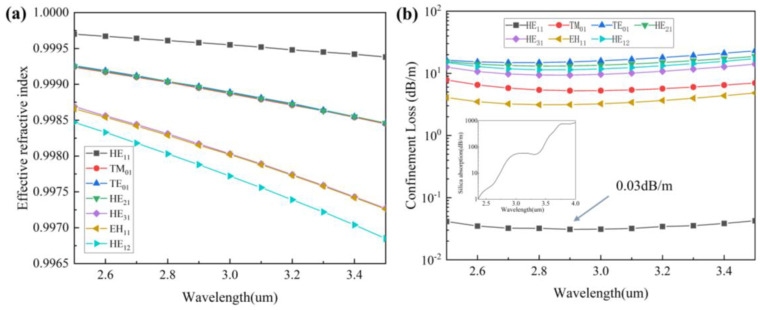
(**a**) Curves of an effective refractive index with wavelengths for different modes. (**b**) Confinement loss of different modes (Insert: loss spectrum of quartz material).

**Figure 5 nanomaterials-13-02440-f005:**
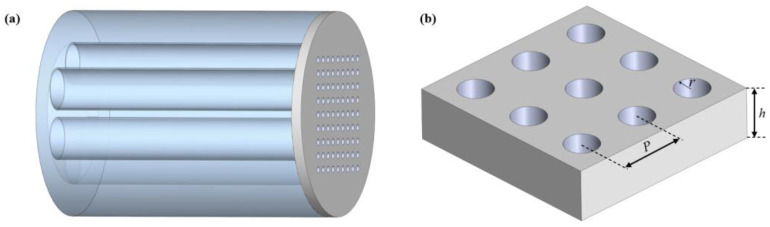
(**a**) Schematic diagram of the fiber-integrated single-layer silicon nitride metasurface device. (**b**) Schematic diagram of a single-layer silicon nitride metasurface filter.

**Figure 6 nanomaterials-13-02440-f006:**
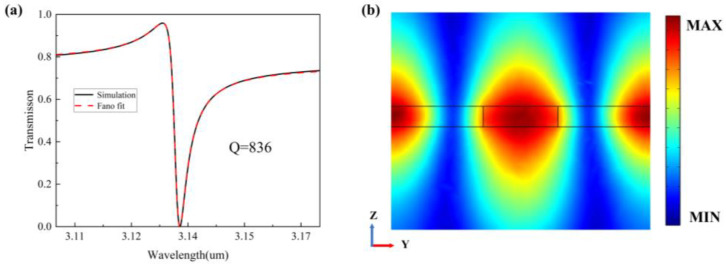
(**a**) Simulation and fitted transmission spectra of the fiber-integrated single-layer silicon nitride metasurface device. (**b**) Mode electric field distribution of a single-layer silicon nitride metasurface filter.

**Figure 7 nanomaterials-13-02440-f007:**
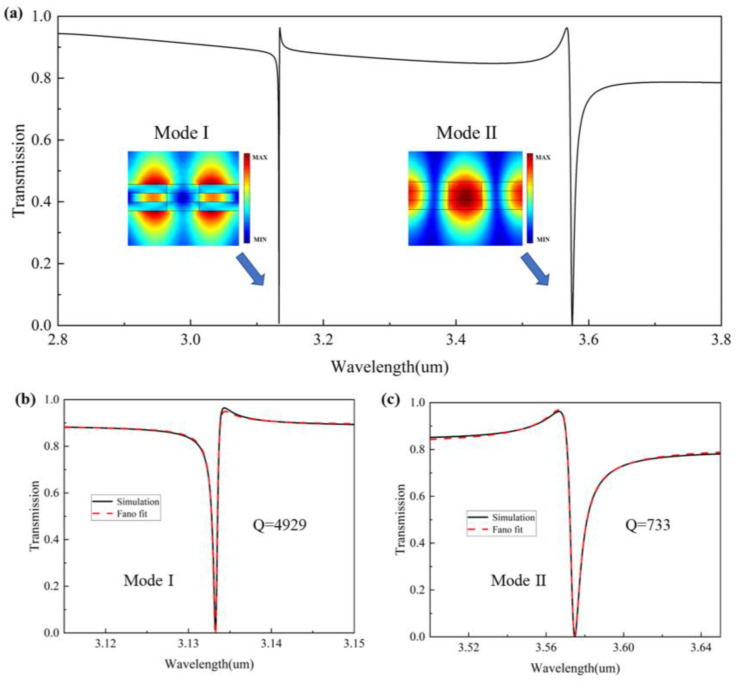
(**a**) Transmission spectra of the fiber-integrated bilayer silicon nitride metasurface device, and simulated mode electric field distribution. (**b**) Simulation and fitted transmission spectra of mode I. (**c**) Simulation and fitted transmission spectra of mode II.

**Figure 8 nanomaterials-13-02440-f008:**
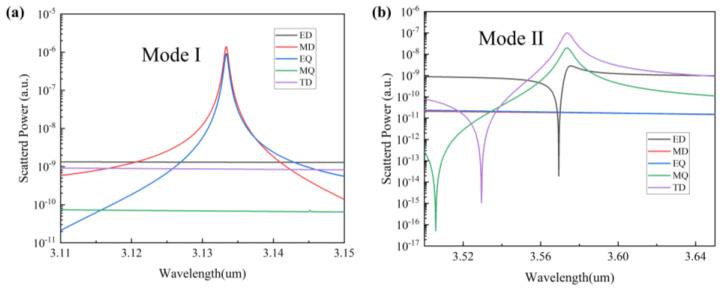
(**a**) Cartesian multipole decomposition of mode I. (**b**) Cartesian multipole decomposition of mode II.

**Figure 9 nanomaterials-13-02440-f009:**
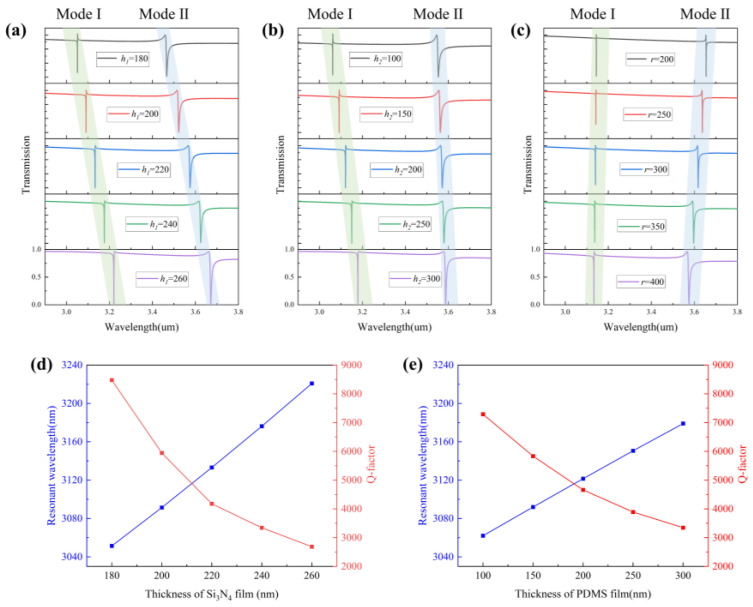
(**a**) Transmittance spectrum under different thicknesses of Si_3_N_4_. (**b**) Transmittance spectrum under the different thicknesses of PDMS. (**c**) Transmittance spectrum under the different radii of air holes. (**d**) The resonant wavelength and Q-factor of mode I as a function of the thickness of the Si_3_N_4_ film. (**e**) The resonant wavelength and Q-factor of mode I as a function of the thickness of the PDMS film.

**Figure 10 nanomaterials-13-02440-f010:**
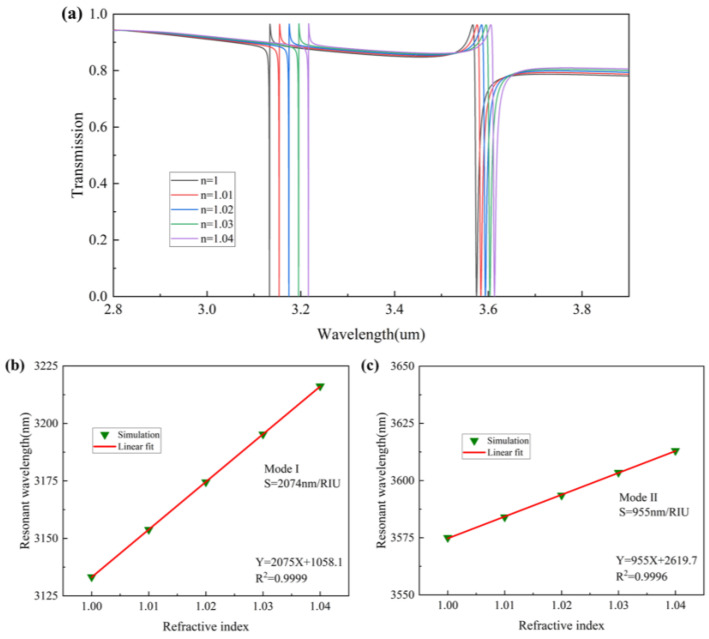
(**a**) Transmission spectra of modes I and II at different background refractive indices. (**b**) Resonant wavelength shifts of mode I versus the refractive index. (**c**) Resonant wavelength shifts of mode II versus the refractive index.

**Figure 11 nanomaterials-13-02440-f011:**
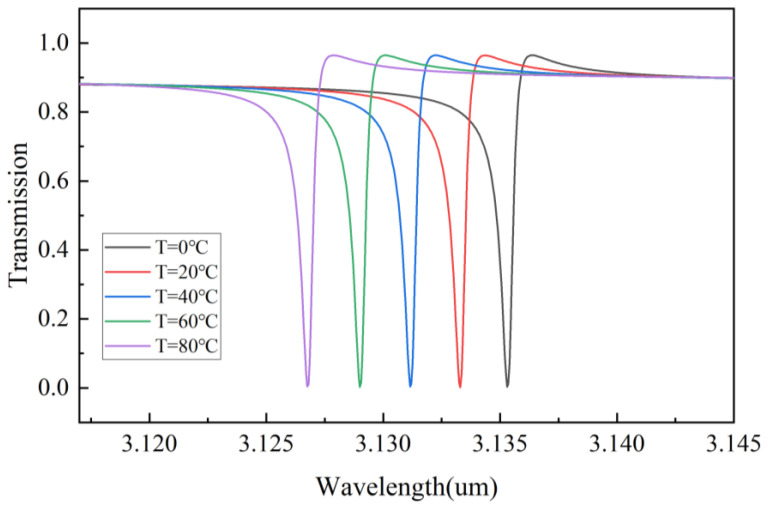
Transmission spectra of mode I at different temperatures.

## Data Availability

Not applicable.

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
