# Peer review of "A Mid-Infrared Multifunctional Optical Device Based on Fiber Integrated Metasurfaces"

_nanomaterials, 2023, doi:10.3390/nano13172440_

Round 1

Reviewer 1 Report

The authors of the paper titled “a mid-infrared multifunctional optical device based on fiber integrated metasurface” numerically demonstrated a fiber-integrated metasurfaces in the mid-infrared wavelength for transmission modulation and refractive index sensing. Even though the concept is clear, and the results are convincing, the manuscript is not well presented. I would recommend a major revision before acceptance. Some of my comments are,

1.     The motivation and the significance of the work is not very clear from the abstract and introduction sections. I would recommend authors to rewrite both sections to make it more interesting for the readers. I also found several typos and some sentences are not clear. What do you mean by “a two-dimensional array of planes that can be tuned to electromagnetism”? What is “raised structural metasurfaces”?

2.     It seems that the proposed fiber-integrated metasurface is a complicated device. The authors should discuss the possible fabrication procedure of the device if someone wants to experimentally validate the presented results.

3.     In Fig. 1c, authors show the schematic of the double-layer silicon nitride metasurface. It seems that the holes are drilled only on the top silicon nitride layer. Then the effective index of the drilled layer must be considered.  If yes, what is the light guiding mechanism of this asymmetric system?   

Moderate editing of English language is required.

Reviewer 2 Report

The authors theoretically proposed a mid-infrared multifunctional optical device based on HC-ARF integrated metasurface for efficient transmission, filtering, and sensing in the mid-infrared band. As a result of a thorough assessment, I conclude that the article requires several significant corrections, so I am asking the authors to thoroughly address the following issues:

1.      In the introduction the authors emphasized that “various metasurfaces were designed on subwavelength dimension cells to modulate the phase, polarization mode, and propagation pattern of electromagnetic waves.” The following works are worth mentioning in this context: Sci Rep 11, 74 (2021), Nanophotonics 12.6 (2023): 1115-1127, etc.

2.     In line 117 the work by Luke et al. 2015 was mentioned. However, there is no reference.

3.     The authors present graphs giving the wavelength in thousands of nanometers. I find it pointless. These values should be given in micrometers. Please correct it.

4.     In Fig 1(c) should be Si3N4 instead SiN.

5.     Unfortunately, theoretical data cannot be reproduced experimentally because the authors did not take into account the dispersion of material parameters. To what extent does the dispersion of refractive indices affect the parameters of the device? Please discuss this thoroughly.

Reviewer 3 Report

A new optical device based on hollow core and integrated metasurface has been proposed, which is potentially find applications for efficient transmission, filtering, and sensing in the mid-infrared technique. The integrated bilayer metasurfaces at the fiber end face can achieve effective modulation of the optical field. The Fano resonance excited by the metasurface can be used to achieve sensing of small variations of the refractive index. The paper contains mostly the results of simulations performed using COMSOL Multiphysics software. As a calculation result, the loss the fiber in the mid-infrared band is found to be extremely low and has good single-mode transmission performance. The relatively high FOM values have been predicted. These high FOM values are potentially valuable for tunable filters and sensors. The paper falls into the scope of the journal. Conclusion of the paper is supported by the text. The list of references is appropriate. The paper can be published after the following insignificant drawbacks will be removed.

1.      In Abstract and in Introduction, where the purposes of the paper is described, the following possibility is announced: “the refractive index of the PDMS can be varied by controlling the temperature, which can adjust the position of the filtered resonant peak”. Further, ”the wavelength of the proposed device can be tuned by changing the temperature of the PCSs”. These statements, however, have not been confirmed in the text of the paper.

2.      The abbreviations PDMS and RIU are not explained.

3.      The angular frequency w  is introduced iteratively in line 286. The position vector g is explained in line 287 after the formulas (10)-(14), but g does not contributing in these formulas.

4.      Please, correct the misprints in lines 74, 90, 112, 238, 242.

The misprints in lines 74, 90, 112, 238, 242 have to be corrected.

Round 2

Reviewer 1 Report

The authors have adequately addressed my comments.

Reviewer 2 Report

The authors followed my suggestions and revised the manuscript. Therefore, I believe that it can be published in its current form.